# Morphological and Molecular Characterization of Anisakid Nematode Larvae (Nematoda: Anisakidae) in the Black Cusk eel *Genypterus maculatus* from the Southeastern Pacific Ocean off Peru

Jhon Darly Chero [1,2,*](ORCID), Luis Ñacari [3,4], Celso Luis Cruces [2,5], David Fermín Lopez [2], Edson Cacique [2](ORCID), Ruperto Severino [2], Jorge Lopez [6], José Luis Luque [7](ORCID) and Gloria Saéz [6]

1   Laboratorio de Zoología, Facultad de Ciencias Biológicas, Universidad Ricardo Palma (URP), Av. Alfredo Benavides 5440 Santiago de Surco, Lima 15039, Peru
2   Laboratorio de Zoología de Invertebrados, Departamento Académico de Zoología, Facultad de Ciencias Biológicas, Universidad Nacional Mayor de San Marcos (UNMSM), Av. Universitaria Cruce con Av. Venezuela Cuadra 34, Lima 15081, Peru; celso.cruces@unmsm.edu.pe (C.L.C.); david.lopez1@unmsm.edu.pe (D.F.L.); edson.cacique@unmsm.edu.pe (E.C.); rseverinol@unmsm.edu.pe (R.S.)
3   Laboratorio de Ecología y Evolución de Parásitos, Facultad de Ciencias del Mar y Recursos Biológicos, Universidad de Antofagasta, 601 Angamos, Antofagasta 1240000, Chile; luis.nacari.enciso@ua.cl
4   Instituto Milenio de Oceanografía, Universidad de Concepción, Concepción 4030000, Chile
5   Programa de Pós-Graduação em Biologia Animal, Universidade Federal Rural do Rio de Janeiro, Seropédica 23890-000, RJ, Brazil
6   Laboratorio de Parasitología General y Especializada, Facultad de Ciencias Naturales y Matemática, Universidad Nacional Federico Villarreal (UNFV), El Agustino, Lima 15007, Peru; jlopezb@unfv.edu.pe (J.L.); gsaez@unfv.edu.pe (G.S.)
7   Departamento de Parasitologia Animal, Universidade Federal Rural do Rio de Janeiro, Seropédica 23890-000, RJ, Brazi; luqueufrrj@gmail.com
*   Correspondence: jhon.chero@urp.edu.pe or jcherod@unmsm.edu.pe

**Abstract:** The back cusk eel, *Genypterus maculatus* (Tschudi, 1846), (Ophiidiformes: Ophiididae) is one of the benthic-demersal fish usually consumed in northern Peru. Here, we identified the third stage (L3) Anisakidae sampled from 29 specimens of *G. maculatus* captured off the south American Pacific coast, Lambayeque Region, Peru. A total of 20 anisakid nematode larvae were collected on the visceral surface and divided morphologically into three types (Type I–III). These larvae were identified by mtDNA Cox2 sequences analysis, which indicated that corresponded to *Anisakis pegreffii* Campana-Rouget and Biocca, 1955, *Skrjabinisakis physeteris* (Baylis, 1923) and *S. brevispiculata* (Dollfus, 1966) Safonova, Voronova, and Vainutis, 2021, respectively. This is the first record of *S. brevispiculata* in Peru. The results obtained in this study provide knowledge on the diversity and distribution of *Anisakis* Dujardin, 1845 and *Skrjabinisakis* Mozgovoi, 1951, species in the south American Pacific waters and their relevance for public health. In addition, we suggest that combined use of molecular and morphological approaches is needed to characterize L3 anisakid larvae.

**Keywords:** *Skrjabinisakis brevispiculata*; *Skrjabinisakis physeteris*; *Anisakis pegreffii*; fish parasites; endoparasites; public health; South America

## 1. Introduction

The back cusk eel, *Genypterus maculatus* (Tschudi, 1846) (Ophidiiformes: Ophidiidae), is one of the benthic-demersal fish species, caught as bycatch in artisanal fisheries, distributed in the Southeastern Pacific in Ecuador, Peru, and Chile (3° S to 53° S) [1–3]. The diet of *G. maculatus* is composed of crustaceans and small teleosts [4]. Inhabits the continental shelf and architectural zone of the slope in sandy–muddy bottoms between 65 and 300 m depth [5].

Nematodes of the family Anisakidae Railliet and Henry, 1912 are cosmopolitan parasites with an indirect life cycle, infecting a wide range of hosts [6,7]. This family includes, among others, the parasitic genera of health relevance, *Anisakis*, *Pseudoterranova*, and *Contracaecum* [8]. The nematode adults infect mainly aquatic mammals and piscivorous birds, while the larval stages are frequently found in aquatic invertebrates (crustaceans and cephalopods) and fishes, which act as intermediate or paratenic hosts [6,9,10].

The presence of anisakid larvae (L3) is enormously important because they cause lesions in fish tissue associated with their mortality and in the fishing industry, causing huge economic losses. Furthermore, they are important pathogens involved in fish-borne zoonotic diseases, which is of a two-fold nature. On the one hand, as a larval invasion, eating live larval forms. On the other hand, as hypersensitivity to thermostable antigens, eating even dead larvae [7,11–14].

For this reason, the taxonomic identification of anisakid larval species is the first step toward the epidemiology and diagnosis of diseases associated with these nematode parasites [7]. However, identification of anisakid larvae at the species level is difficult using only morphological data due to the low development of the organs and the absence of diagnostic characters, which appear in the adult nematodes [7]. Consequently, for more precise identification in the larval phase, molecular techniques started to be used [15–17].

In Peru, anisakid nematode larvae (L3) have been recovered from the visceral surface of several marine teleost fishes and of a cephalopod species [18]. Four anisakid larvae were reported, *Anisakis simplex* (Rudolphi, 1809), *Anisakis physeteris* (Baylis, 1923), *Contracaecum multipapillatum* (Drasche, 1882), and *Pseudoterranova decipiens* (Krabbe, 1878). Specific identification of these larvae was based only on morphological data. Recent molecular studies indicate the presence of *Anisakis pegreffii*, infecting commercial fish from the Peruvian coast [19,20].

Due to the scarcity of data in northern Peru on the diversity of anisakid larval species and their possible risk to human health, the aim of the present work was to identify anisakid nematodes found in the black cusk eel *Genypterus maculatus*, a popular fish in local markets from northern Peru, using combined molecular (mitochondrial cytochrome c-oxidase subunit II) and morphological (light and scanning electron microscopy) approaches.

## 2. Materials and Methods

### 2.1. Specimen Collection and Morphological Analyses

Twenty-nine specimens of *G. maculatus* were caught by local fishermen with a fishing line from off the coastal zone of Puerto Santa Rosa, Lambayeque Region, Peru (6°52′ S, 79°55′ W), between March 2022 and July 2022 (autumn-winter). All fish were weighed, measured, and subsequently necropsied. Fish nomenclature and classification follow Froese and Pauly (2021). Nematodes were removed from the host visceral surface, washed in physiological saline, fixed in hot 70% ethanol, and preserved in 90% ethanol until use. The anterior and posterior parts of each nematode larvae were cut and used for morphological identification, while the middle parts were used in molecular procedures. Nematodes were cleared in lactophenol for observations and measurements under light microscopy. Specimens were examined using a compound Nikon™ Eclipse SI photomicroscope (Tokyo, Japan) equipped with phase contrast microscopy optics and drawings were made with the aid of a drawing tube. Unless stated otherwise, measurements are in micrometers, representing straight-line distances between extreme points of the structures measured and are expressed as the range followed by the mean and number (*n*) of structures measured in parentheses. Some nematodes were taken for scanning electron microscopy (SEM), dehydrated through a graded ethanol series, critical point dried with carbon dioxide, coated with gold, and examined in an Inspect S50—FEI, at an accelerating voltage of 7 kV. The prevalence of anisakid parasites was calculated according to Bush et al. [21]. A voucher specimen was deposited in the Helminthological Collection in the Museum of Natural History at the San Marcos University (MUSM-HEL), Lima, Peru.

### 2.2. DNA Extraction, PCR Amplification and DNA Sequencing

The middle parts of nematodes were prepared for total genomic DNA extraction using a Genomic DNA Mini Tissue Kit (Geneaid Biotech Ltd., New Taipei City, Taiwan), according to the manufacturer's instructions. The mitochondrial cytochrome c oxidase subunit II gene (mtDNA cox2) was amplified using the primers 210 (5′-CACCAACTCTTAAAATTATC-3′) and 211 (5′-TTTTCTAGTTATATAGATTGRTTYAT-3′) [22]. PCR reactions were performed according to Martinez-Rojas et al. [20]. PCR products were visualized with Sybergreen (Invitrogen, Eugene, Oregon, EUA) staining before electrophoresis on 1.5% agarose gels. The amplified PCR products were purified with GenepHlow Gel/PCR Kit (Geneaid Biotech Ltd., New Taipei City, Taiwan), following the manufacturer's instructions and sequenced in Bio Basic Inc. (Markham city, Canada) with the Sanger sequencing method. Sequences were edited and contigs were assembled using ProSeq 2.9 beta [23]. The National Center for Biotechnology Information (NCBI) sequence database (henceforth 'GenBank') was searched for similar sequences using BLAST (Basic Local Alignment Search Tool) [24].

### 2.3. Molecular Analyses

Sequences generated in this study were aligned with selected sequences obtained from GenBank, using the software Clustal W (Table 1) [25]. *Hysterothylacium aduncum* (Rudolphi, 1802) (GenBank: JQ934891) was set as an outgroup for the *cox2* phylogenetic analysis. The aligned dataset was analyzed with the software JModelTest2 [26]. The best model found by JModelTest2, selected with the corrected Akaike information criterion [27], was TIM1 + G. The model parameters were as follows: assumed nucleotide frequencies A = 0.2179, C = 0.0921, G = 0.2325, and T = 0.4575; substitution rate matrix with A-C substitution = 1.0000, A-G= 8.0808, A-T = 0.4611, C g = 0.4611, C-T = 15.7526, G-T = 1.000, and gamma distribution with shape parameter 0.1750. Next, the best model was implemented in MrBayes 3.2.7a [28] for Bayesian Inference analysis (BI) and in IQ-TREE [29] for Maximum Likelihood analysis (ML). All phylogenetic analyses were conducted in the CIPRES Science Gateway V. 3.3 platform (http://www.phylo.org/ (accessed on 22 April 2023)) [30].

For the BI analysis, unique random starting trees were used in the Metropolis-coupled MCMC [28]. The analysis was performed for a total of 5,000,000 generations. Visual inspection of log-likelihood scores against generation time indicated that the log-likelihood values reached a stable equilibrium before the 100,000th generation. Thus, a burn-in of 1000 samples was conducted; every 100th tree was sampled from the MCMC analysis, obtaining a total of 100,000 trees and tree topology represented the 50% majority rule consensus trees. Support for nodes in the BI tree topology was obtained by posterior probability. For the ML analysis, we used the default options in IQ-TREE run through the Cypress Science Gateway [29]. The robustness of the ML tree topology was assessed by bootstrap iterations of the observed data 1000 times. Phylogenetic trees were visualized and edited in Figtree 1.4.4. Pairwise genetic distances (intra and interspecific) between the sequences of cox1 gene were calculated in MEGA [31] using the Kimura 2-Parameter model [32].

**Table 1.** Specimen information and GenBank accession numbers in on mtDNA cox2 gene. Stage: A = Adult; L = Larvae. Sequences obtained for the present study are in bold.

| Access | Species | Host | Country | Stage | Reference |
|---|---|---|---|---|---|
| DQ116432 | *Skrjabinisakis physeteris* | *Physeter macrocephalus* | Mediterranean Sea | A | [33] |
| AB592801 | *Skrjabinisakis physeteris* | *Beryx splendens* | Japan | L | [34] |
| **OR192868** | ***Skrjabinisakis physeteris* (Sphy1)** | ***Genypterus maculatus*** | **Southeastern Pacific Ocean** | **L** | **Present study** |
| **OR192869** | ***Skrjabinisakis physeteris* (Sphy2)** | ***Genypterus maculatus*** | **Southeastern Pacific Ocean** | **L** | **Present study** |
| **OR192870** | ***Skrjabinisakis physeteris* (Sphy3)** | ***Genypterus maculatus*** | **Southeastern Pacific Ocean** | **L** | **Present study** |
| **OR192871** | ***Skrjabinisakis physeteris* (Sphy4)** | ***Genypterus maculatus*** | **Southeastern Pacific Ocean** | **L** | **Present study** |
| **OR192872** | ***Skrjabinisakis physeteris* (Sphy5)** | ***Genypterus maculatus*** | **Southeastern Pacific Ocean** | **L** | **Present study** |
| **OR192873** | ***Skrjabinisakis physeteris* (Sphy6)** | ***Genypterus maculatus*** | **Southeastern Pacific Ocean** | **L** | **Present study** |
| MH669506 | *Skrjabinisakis brevispiculata* | *Diaphus* sp. | Indian Ocean | L | [35] |
| DQ116433 | *Skrjabinisakis brevispiculata* | *Kogia breviceps* | No registred | A | [33] |
| **OR192874** | ***Skrjabinisakis brevispiculata* (Sbre1)** | ***Genypterus maculatus*** | **Southeastern Pacific Ocean** | **L** | **Present study** |
| **OR192875** | ***Skrjabinisakis brevispiculata* (Sbre2)** | ***Genypterus maculatus*** | **Southeastern Pacific Ocean** | **L** | **Present study** |
| **OR192876** | ***Skrjabinisakis brevispiculata* (Sbre3)** | ***Genypterus maculatus*** | **Southeastern Pacific Ocean** | **L** | **Present study** |
| **OR192877** | ***Skrjabinisakis brevispiculata* (Sbre4)** | ***Genypterus maculatus*** | **Southeastern Pacific Ocean** | **L** | **Present study** |

**Table 1.** *Cont.*

| Access | Species | Host | Country | Stage | Reference |
|---|---|---|---|---|---|
| DQ116434 | *Skrjabinisakis paggiae* | *Kogia breviceps* | West Atlantic Ocean (Florida coast) | A | [33] |
| AB592807 | *Skrjabinisakis paggiae* | *Beryx splendens* | Japan | L | [34] |
| DQ116430 | *Anisakis ziphidarum* | *Mesoplodon layardii* | Southeast Atlantic Ocean (South African coast) | A | [33] |
| AB517573 | *Anisakis ziphidarum* | *Scomber japonicus* | Japan | L | [36] |
| DQ116431 | *Anisakis nascetti* | *Mesoplodon miros* | Southeast Atlantic Ocean (South African coast) | A | [33] |
| GQ118167 | *Anisakis nascetti* | *Mesoplodon grayi* | From off New Zealand | A | [37] |
| DQ116427 | *Anisakis typica* | *Delphinidae* | Western North Atlantic Ocean | A | [33] |
| KC928266 | *Anisakis typica* | *Katsuwonus pelamis* | Southern Makassar Strait, Indonesia | L | [38] |
| KC810003 | *Anisakis simplex* | *Balaenoptera acutorostrata* | Northeastern Atlantic Ocean (Norwegian coast) | A | [39] |
| DQ116426 | *Anisakis simplex* | Delphinidae | Northeast Pacific coast | A | [33] |
| MZ546440 | *Anisakis pegreffii* | *Seriolella violacea* | Southeastern Pacific Ocean | L | [20] |
| DQ116428 | *Anisakis pegreffii* | *Delphinus delphis* | Northeast Atlantic Ocean (Spanish coast) | A | [33] |
| **OR192866** | ***Anisakis pegreffii* (Apeg1)** | ***Genypterus maculatus*** | **Southeastern Pacific Ocean** | **L** | **Present study** |
| **OR192867** | ***Anisakis pegreffii* (Apeg2)** | ***Genypterus maculatus*** | **Southeastern Pacific Ocean** | **L** | **Present study** |
| MN385245 | *Anisakis berlandi* | *Globicephala melas* | New Zealand | A | [40] |
| KC810000 | *Anisakis berlandi* | *Globicephala melas* | New Zealand | A | [39] |
| JQ934891 | *Hysterothylacium aduncum* * | *Trachurus trachurus* | Croatia | L | [41] |

* Species used as outgroup.

## 3. Results

A total of 20 out of 29 (68.9%) *G. maculatus* revealed the presence of anisakid nematode larvae in the coelomic cavity. All larvae were mainly found encysted on the surface of the liver and intestines. A total of 20 anisakid nematode larvae were collected and assigned morphologically to the genus *Anisakis* into three types (Type I–III). Type I larvae were characterized by having an elongated ventricle and a mucron at the posterior end, type II larvae had a short ventricule and no mucron, and type III larvae had a short ventricule and a short tail and lacking mucron. According to the sequence analysis at the mtDNA Cox2 gene locus, larvae (type I–III) were identified as *Anisakis pegreffii* by Campana-Rouget, and Biocca, 1955, *Skrjabinisakis physeteris* (Baylis, 1923) by Safonova, Voronova and Vainutis, 2021, and *S. brevispiculata* (Dollfus, 1966) by Safonova, Voronova, and Vainutis, 2021, respectively.

### 3.1. Systematics and Morphological Characteristics of the Anisakid Larvae

- Class Chromadorea Inglis, 1983.
- Order Rhabditida Chitwood, 1933.
- Anisakidae Railliet and Henry, 1912.

### 3.1.1. *Anisakis pegreffii* Campana-Rouget and Biocca, 1955 (Figures 1A,B and 2A,B)

The host is *Genypterus maculatus* (Tschudi, 1846) (Ophidiiformes: Ophidiidae), a back cusk-eel. The locality is off the coastal zone of Puerto Santa Rosa (6°52′ S, 79°55′ W), Lambayeque Region, northern Peru.

- Site in host: body cavity.
- Specimens deposited: Hologenophore (MUSM-HEL 5141).
- Representative DNA sequence: Sequences were deposited in GenBank under the accession numbers OR192866 and OR192867 for the mtDNA cox2.

The description is based on one specimen: a third-stage larvae. It has a slender body, is cylindrical, and 20 mm long, with a fine transversal and longitudinal cuticular striations along the body, which are more evident on anterior and posterior ends. The cephalic end is rounded, bearing a small cuticular larval tooth. The mouth surrounded by one dorsal and two ventrolateral lips; the lips are poorly developed; the dorsal lip has two indistinct cephalic papillae; the ventrolateral lips each have one cephalic papillae. It also has a triangular oral opening. The pore excretory is situated below the dorsal lip base. The nerve ring 202 is on the anterior end. The esophagus is 2.7 mm long and 293 mm wide, representing 13.5% of total body length. The ventriculus is long, dolioform, 1.12 mm long, and 400 mm wide, representing 41.45% of the esophagus length. The rectum short hyaline tube with two unicellular rectal glands. The tail is short, rounded, and 139 mm long, with a terminal cylindrical bentley protruded mucron; the body length/tail length is 143.8 mm; the mucron is 26 mm long.

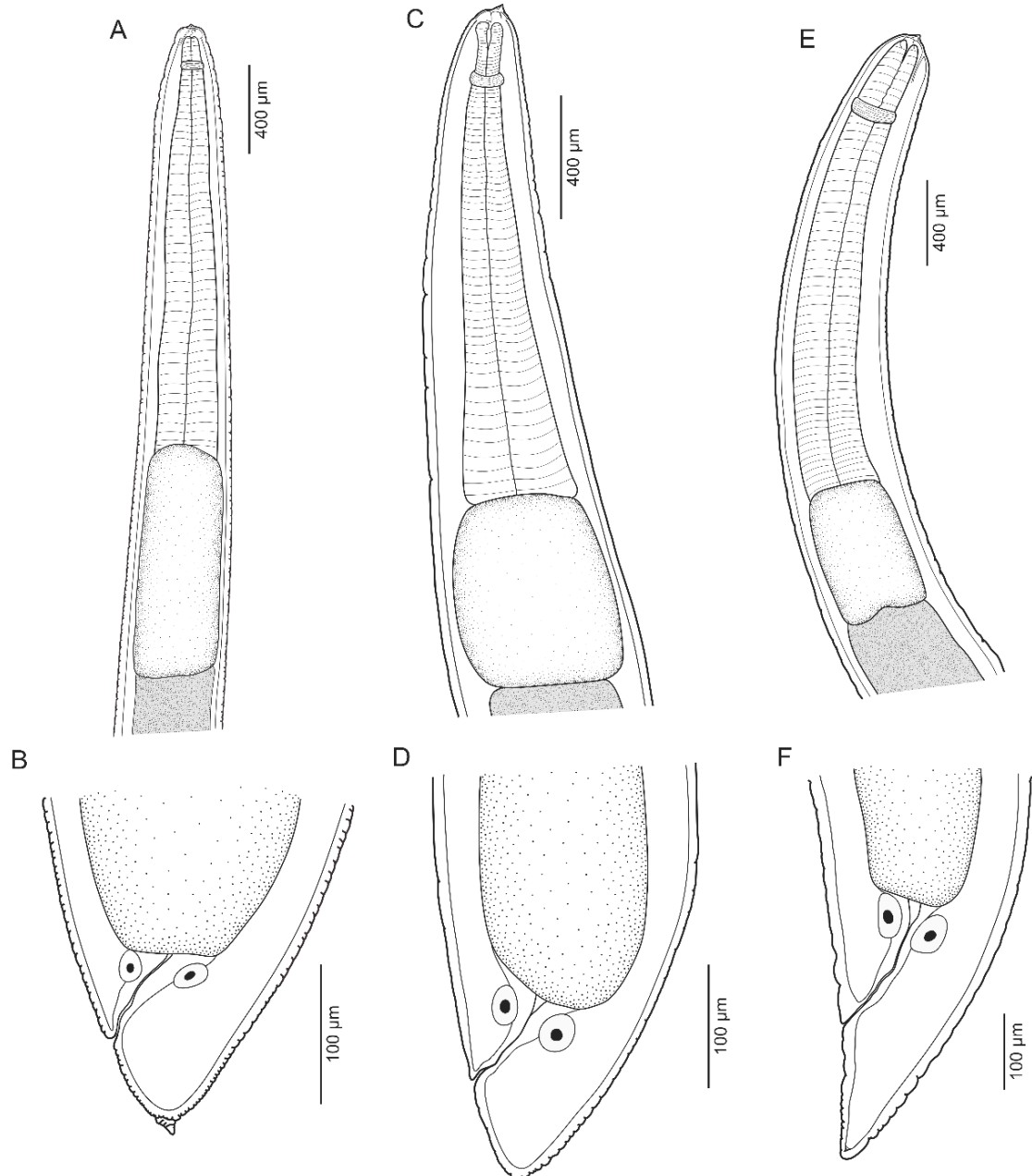

**Figure 1.** Morphology of the isolated larvae from *Genypterus maculatus*. (**A**) Anterior end of *A. pegreffi*, (**B**) Posterior end of *A. pegreffi*, (**C**) Anterior end of *S. brevispiculata*, (**D**) Posterior end of *S. brevispiculata*, (**E**) Anterior end of *S. physeteris*, and (**F**) Posterior end of *S. physeteris*.

3.1.2. *Skrjabinisakis brevispiculata* (Dollfus, 1966) Safonova, Voronova and Vainutis, 2021 (Figures 1C,D and 2C,D)

The host is *Genypterus maculatus* (Tschudi, 1846) (Ophidiiformes: Ophidiidae), a back cusk-eel. The locality is off the coastal zone of Puerto Santa Rosa (6°52′ S, 79°55′ W), Lambayeque Region, northern Peru.

- Site in host: body cavity.
- Specimens deposited: Hologenophore (MUSM-HEL 5142).
- Representative DNA sequence: Sequences were deposited in GenBank under the accession numbers OR192874–OR192877 for the mtDNA cox2.

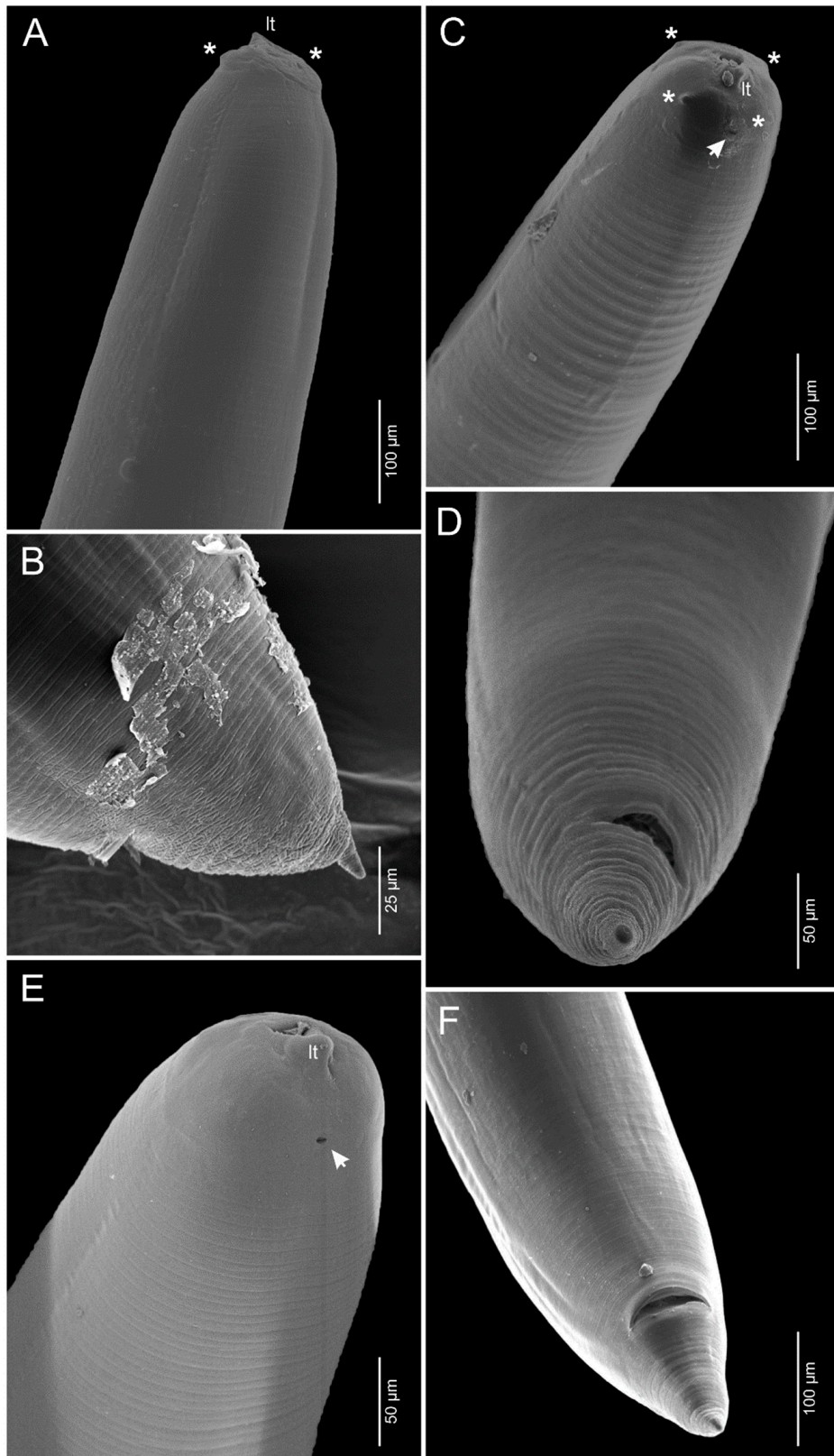

**Figure 2.** Scanning electron microscopy images of the isolated larvae from *Genypterus maculatus*. (**A**) Anterior end of *A. pegreffi*; (**B**) Posterior end of *A. pegreffi*; (**C**) Anterior end of *S. brevispiculata*; (**D**) Posterior end of *S. brevispiculata*; (**E**) Anterior end of *S. physeteris*; (**F**) Posterior end of *S. physeteris*. Asterisks indicate the cephalic papillae and the arrow indicates the excretory pore. Lt: larval tooth.

The description based on three specimens: the third-stage larvae. The body is slender, cylindrical, and 20–27 (24) mm long, with fine transversal and longitudinal cuticular striations along the body, which are more evident on the anterior and posterior ends. The cephalic end is rounded, bearing a small cuticular larval tooth. The mouth is surrounded by one dorsal and two ventrolateral lips; the lips poorly developed; the dorsal lip has two cephalic papillae; the ventrolateral lips each have one cephalic papillae. There is a triangular oral opening. The pore excretory is situated below the dorsal lip base. The nerve ring 272–320 (299) is from anterior end. The esophagus is 1.5–1.7 (1.6) mm long and 216–556 (333) mm wide, representing 6.15–7.5 (6.65)% of total body length. The ventriculus is short, oblong, 422–665 (544) mm long, and 340–560 (440) mm wide, representing 28.13–39.11 (33.79)% of the esophagus length. The rectum short hyaline tube has two unicellular rectal glands. The tail is conical, short, 98–113 (106) mm long and the body length/tail length is 176.9–275.5 (226.2) mm.

### 3.1.3. *Skrjabinisakis physeteris* (Baylis, 1923) Safonova, Voronova and Vainutis, 2021 (Figures 1E,F and 2E,F)

The host is *Genypterus maculatus* (Tschudi, 1846) (Ophidiiformes: Ophidiidae), a back cusk-eel. The locality is off the coastal zone of Puerto Santa Rosa (6°52′ S, 79°55′ W), Lambayeque Region, northern Peru.

- Site in host: body cavity.
- Specimens deposited: Hologenophore (MUSM-HEL 5143).
- Representative DNA sequence: Sequences were deposited in GenBank under the accession numbers OR192868–OR192873 for the mtDNA cox2.

The description is based on two specimens third-stage larvae. The body is slender, cylindrical, and 32–34 (33) mm long, with fine transversal and longitudinal cuticular striations along body, which are more evident on the anterior and posterior ends. The cephalic end is rounded, bearing a small cuticular larval tooth. The mouth is surrounded by one dorsal and two ventrolateral lips; the lips are poorly developed; the dorsal lip has two cephalic papillae; the ventrolateral lips each have one cephalic papillae. There is a triangular oral opening. The pore excretory is situated below the dorsal lip base. The nerve ring is 305–362 (334) mm from the anterior end. The esophagus is 2.1–5 (3.5) mm long and 276–301 (289) mm wide, representing 6.17–15.62 (10.09)% of total body length. The ventriculus is short, oblong, 620–689 (654) mm long, and 349–406 (378) mm wide, representing 12.4–32.80 (22.60)% of the esophagus length. The rectum has a short hyaline tube with two unicellular rectal glands. The tail is elongated and 211–285 (248) mm long; the body length/tail length is 112.3–161.1 (136.7) mm.

### 3.2. *Phylogenetic Analyses*

The mtDNA *cox2* sequences were determined for 3 anisakid nematodes isolated from the 20 infected *G. maculatus*. The phylogenetic analyses included 31 *cox2* sequences, 550 bp in length (after alignment): 12 sequences were obtained during this study and 19 sequences were retrieved from GenBank (Table 1). The data matrix comprised a total of 158 parsimony informative sites.

In the BI and ML phylogenetic trees, the genus *Skrjabinisakis* formed a clade, with *S. physeteris* and *S. brevispiculata* being sister taxa forming a clade with that of *S. paggiae* and bootstrap values at the two branch points of three reference strains were 100% (BI) and 97% (ML) (Figures 3 and 4). The six sequences larvae (Sphy1-Sphy6) clustered with the adults of *Skrjabinisakis physeteris* of *Physeter macrocephalus* in the Mediterranean Sea (AB592801) and larvae of *Beryx splendens* in Japan (DQ116432) (Figures 3 and 4). The sequences of larvae (Sbre1-Sbre4) clustered with an adult of *S. brevisculata* of *Kogia breviceps* is not located in a register (DQ116433) and larvae of *Diaphus* sp. In the Indian Ocean (MH669506) (Figures 3 and 4) and two sequences of larvae (Apeg1 y Apeg2) clustered with adult *Anisakis pegreffii* of *Delphinus delphis* in Northeastern Atlantic Ocean (DQ116428) and larvae of the *Seriolella violacea* on the Peruvian coast (Figures 3 and 4).

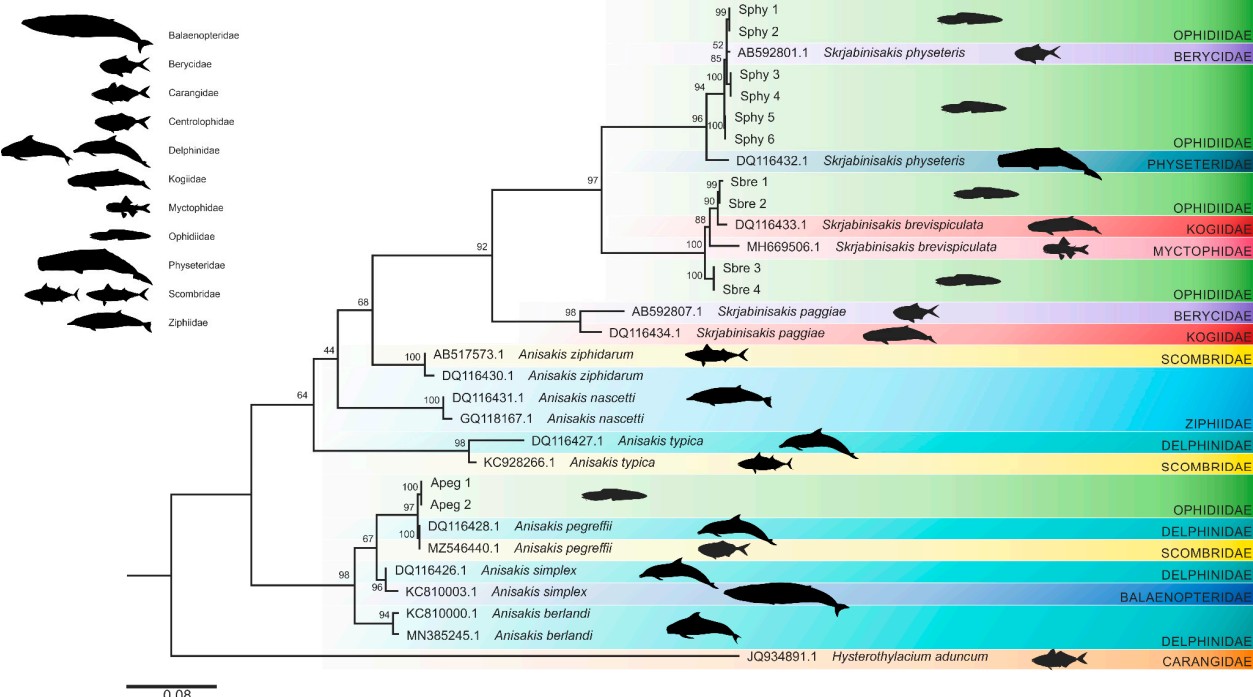

**Figure 3.** Maximum likelihood tree of the isolated anisakid larvae from *Genypterus maculatus* based on the mtDNA cox2 to show their relationships with other anisakid species. Numbers (%) on the branches indicate 5000 bootstrap replicates. The scale bar represents the number of substitutions per site.

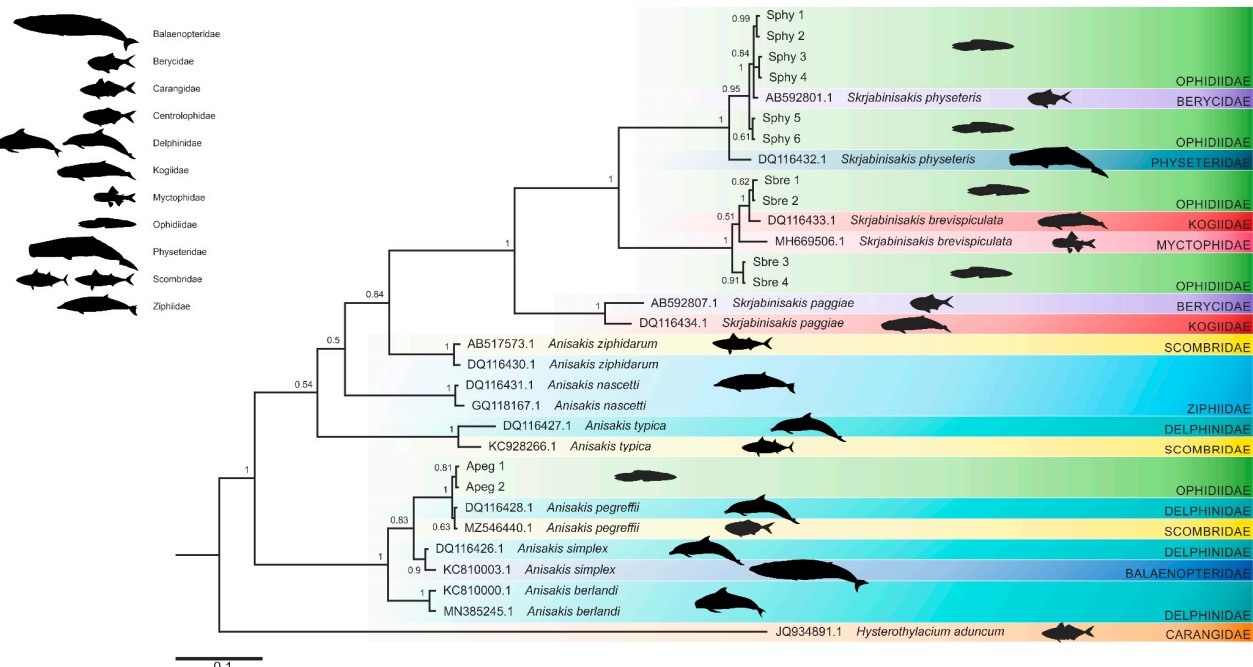

**Figure 4.** Bayesian consensus phylogenetic tree based on the mtDNA cox2 of the isolated anisakid larvae from *Genypterus maculatus* to show their relationships with other anisakid species. The numbers along the branches indicate the bootstrap values obtained from the posterior probability of BI. The scale bar represents the number of substitutions per site.

Distances were computed to the Kimura-2 parameter (K2P) and the number of bp pairwise differences (Table S1_Supplementary material). The distance between the sequence (Sphy1-Sphy6) with *S. physeteris* (AB592801) was 0.37–0.57% (2–3 bp pairwise) and

*S. physeteris* (DQ116432) was 3.31–4.02% (17–18 bp pairwise). The distance between the sequence (Sbre1-Sbre4) with *S. brevisculata* (DQ116433) was 1.16–2.74% (5–13 bp pairwise) and *S. brevisculata* (MH669506) was 3.44–3.40% (16–19 bp pairwise). The distance between the sequence (Apeg 1 and Apeg2) with *Anisakis pegreffii* (DQ116428) was 0.37% (2 bp pairwise).

## 4. Discussion

The present survey is the first record of the identification of anisakid larvae from *G. maculatus* captured off northern Peru, using combined molecular (the mtDNA cox2) and morphological (light and scanning electron microscopy) approaches. We have confirmed the presence of three anisakid nematode species (*A. pegreffi*, *S. physeteris*, and *S. brevispiculata*), infecting *G. maculatus*, a commercially important fish off the coast of the southeast Pacific [1,42].

Morphological identification of *Anisakis* larvae at the species level is difficult due to the absence of taxonomic characteristics. However, Berland [43] classified *Anisakis* larvae into two types, namely *Anisakis* types I and II, based on the length of the ventriculus and the presence or absence of a mucron at the tip of the tail. Subsequently, Murata et al. [34] found other taxonomic criteria (the ratio tail length/body length) to discriminate between three species of the type II larval species complex. Recently, Cabrera-Gil et al. [35] simplified this information, indicating that type II larvae have three subtypes (*Skrjabinisakis physeteris*, *S. brevispiculata*, and *S. paggiae*). Following the criteria of Cabrera-Gil et al. [35], in this study, we classify the three nematode larvae as *Anisakis pegreffi* (Type I), *Skrjabinisakis physeteris* (Type II, subtype 2), and *S. brevispiculata* (Type II, subtype 3). According to Cabrera-Gil et al. [35], type II larvae (*S. physeteris*) have a slightly tilted or parallel tooth and had a long, conical, and tapering tail without a mucron, while type III (*S. brevispiculata*) and type IV larvae (*S. paggiae*) have only a slightly tilted tooth. In addition, type III larvae have a short and rounded tails, some larvae with a tiny spine-like mucron, and type IV larvae have short, conical, and pointed tails without a mucron.

Our phylogenetic analysis (ML and BI), obtained from the mtDNA cox2 sequences, suggests the clade formed by *S. physeteris*, *S. brevispiculata*, and *S. paggiae*, species which until most recently belonged to the genus *Anisakis*, according to Safonova et al. [44] (Figures 2 and 3). Safonova et al. [44] proposed the resurrected generic status of *Peritrachelius* for *A. typica*, and the use of *Skrjabinisakis* as a genus name rather than a subgenus for *A. brevispiculata*, *A. paggiae*, and *A. physeteris* based on the intraspecific genetic distances of ITS sequences. These observations were also indicated by Takano and Sata [45] and Bao et al. [46], who use multiple genetic markers and indicate the validity of the genus *Skrjabinisakis*. Our results also provide conclusive proof of the validity of the genus *Skrjabinisakis*. However, we hesitate to assign *A. typica* to *Peritrachelius* [44], given that the species was nested in *Anisakis* s.s. in the present phylogenetic trees (Figures 2 and 3). *Anisakis* s.s. and *Skrjabinisakis* species are distinguished morphologically, at the adult stage, from each other principally by longer and thinner male spicules and a longer ventriculus of the former [44]. In the larval stage, species of both genera are differentiated mainly by the size of the ventriculus (shorter ventriculus in *Skrjabinisakis* larvae vs. longer ventriculus in *Anisakis* larvae) [45].

*A. pegreffi* is distributed in the Mediterranean Sea and the Austral region between 30° N and 55° S [11]. The larval stages of this species parasitize various teleost fishes and adult nematodes are found infecting delphinids [47]. In the Southeast Pacific, *A. pegreffi* is reported, based on morphological and molecular analyses, parasitizing commercial fish species, i.e., *Trachurus murphyi*, *Merluccius gayi*, *Scomber japonicus*, and *Seriolela violacea* [19,20]. In this study, *G. maculutus* is a new host record to *A. pegreffi* in the Southeast Pacific.

The other two larval nematode parasites found in the present study are *S. physeteris* and *S. brevispiculata*, both have been previously reported to infect sperm whales and fishes [35,48]. *Skrjabinisakis physeteris* is recorded in the Mediterranean, Pacific, and Atlantic Oceans [11,20,34], and *S. brevispiculata* is recorded in the South and Central Atlantic Ocean and in the Pacific on the coast of Japan [35]. To date, the only record of a *Skrjabinisakis*



species in the Southeast Pacific was performed by Martínez-Rojas et al. [20], who found that *S. physeteris* parasitizes *S. japonicus* [20]. Thus, the present work represents the first report of *S. brevispiculata* in the Southeast Pacific and *G. maculutus* is considered a new host record to *S. brevispiculata* and *S. physeteris*.

The host *G. maculatus* is a demersal species, which inhabits the rocky shelf and upper slope waters (50–500 m in depth) [1]. According to Bahamonde and Zavala [4], the diet of *G. maculatus* is principally based on stomatopods and decapods and in a small percentage of squids, sardines, anchovies, and merluccid hakes, which could play a role as paratenic/intermediate hosts in the life cycle of the genus *Anisakis* and *Skrjabinisakis* in the Southeastern Pacific Ocean.

It is interesting to highlight the fact that only two studies, using morphological and molecular analysis, showed the co-occurrence of three anisakid larvae in the same host [34,49]. Quiazon et al. [49] reported the species *A. pegreffii*, *A. simplex*, and *S. brevisculata* from *Gadus chalcogrammus* Pallas, 1814 (Gadidae). Murata et al. [34] registered *S. physeteris*, *S. brevispiculata*, and *S. paggiae*, parasitizing *Beryx splendens* Lowe, 1834 (Berycidae). Similarly, in our study, three Anisakidae species were found to infect the same host species. Therefore, the combined use of molecular and morphological approaches is needed to characterize the L3 anisakid larvae. Of the Anisakidae species found in this study, only *A. pegreffii* have been reported as the causative agent of infection in humans, whereas the pathogenic factor is thermostable proteins of the larval origin that cause hypersensitivity reactions (human food fish poisoning) [19,20]. Finally, the present records provide us with valuable information about the presence of anisakid species in the Southeastern Pacific Ocean.

## 5. Conclusions

In the present study, three Anisakidae species were identified using morphological and molecular analysis. We suggest that the combined use of molecular and morphological approaches is needed to characterize the L3 anisakid larvae. *Skrjabinisakis brevispiculata* is recorded for the first time in Peru.

**Supplementary Materials:** The following supporting information can be downloaded at https://www.mdpi.com/article/10.3390/d15070820/s1, Table S1: Pairwise sequence divergences for mtDNA cox2 sequences among species of *Anisakis* and *Skrjabinisakis*. The Kimura-2-parameter (Kimura 1980, K2P) distances are shown as percentages (below the diagonal) and the raw number of bp-pairwise differences above the diagonal.

**Author Contributions:** J.D.C., C.L.C., L.Ñ., G.S., J.L. and J.L.L. conceived and designed the study; J.D.C., D.F.L., E.C. and G.S. carried out the field work; J.D.C., C.L.C. and L.Ñ. performed molecular analyses. Additional analyses were performed by J.D.C., C.L.C., L.Ñ., D.F.L., E.C. and R.S.; J.D.C. and L.Ñ. wrote the manuscript. All authors have read and agreed to the published version of the manuscript.

**Funding:** C.L.C. was supported by a student fellowship from the Coordenação de Aperfeiçoamento de Pessoal do Ensino Superior, Brazil (CAPES)—Finance Code 001. J.L.L. was supported by a Researcher fellowship from the Conselho Nacional de Desenvolvimento Científico e Tecnológico, Brazil (CNPq).

**Institutional Review Board Statement:** This study did not consider experiments with live animals. All fishes were obtained from commercial catches and none of the species are subject to conservation measures.

**Informed Consent Statement:** Not applicable.

**Data Availability Statement:** Data are available as Supplementary Materials.

**Acknowledgments:** The authors are grateful to the following people who helped to the collection of fishes in Peru: Luis Santillán, Sergio Santillán, and Nathaly Daga, all from the National University of San Marcos (UNMSM).

**Conflicts of Interest:** The authors declare no conflict of interest.

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
