# Peer review of "Morphological and Molecular Characterization of Anisakid Nematode Larvae (Nematoda: Anisakidae) in the Black Cusk eel Genypterus maculatus from the Southeastern Pacific Ocean off Peru"

_diversity, doi:10.3390/d15070820_

Round 1
Reviewer 1 Report
The manuscript is a quality well-written description of study of J3 anisakid nematodes with using both morphological and molecular genetic methods. The study is carried out in accordance with all the rules of similar works. It should be noted original figures that improve the perception of the article. I read the manuscript with great interest. I congratulate the authors for the extensive effort made in carrying out a high-quality scientific paper.
However, I have some comments, which will undoubtedly improve the article.
The article needs traditional Сonclusion.
Please, insert your wonderful figures immediately after their first mention in the text.
Subsections in section 2 Materials and methods need to be numbered – 2.1., 2.2., 2.3.
And all words in subsection headings must be capitalized as well as the article name.
Check that all sections of the article are aligned in width (see Molecular analyses, Phylogenetic analyses).
Line 64 – Indent.
Lines 31,32 – Of course, the merits of the authors who re-described or moved the species to another genus are great, but recent rules must be followed. It is enough to cite the first author (authors) who described the species, like this: Skrjabinisakis physeteris (Baylis, 1923), S. brevispiculata (Dollfus, 1966).
Line 118 – First mention of species in the text. According the International Code of Zoological Nomenclature, its full Latin name with the author and year of description should be given – Hysterothylacium aduncum (Rudolphi, 1802).
Line 144 – Table 1
Line 25, 27, 41, 149, 177, 197, 217, 263, 314 – back cusk-eels ?
Better use Latin name – G. maculatus. Try to avoid common names in scientific articles or must be used together with the Latin name – the black cusk-eel Genypterus maculatus.
I hope that by the time the article is accepted for publication, the accession numbers in Genbank will appear?
Please, consider using “juveniles” instead of “larvae”
Line 160 – The subsection name should be changed to “Systematics and Morphological Characteristics of the Anisakid Juveniles”. Please, rearrange parts of subsection 3.1 in this way:
3.1. Systematics and Morphological Characteristics of the Anisakid Juveniles
Class Chromadorea Inglis, 1983
Order Rhabditida Chitwood, 1933
Anisakidae Railliet & Henry, 1912
Anisakis pegreffii Campana-Rouget & Biocca, 1955
Host: the black cusk-eel Genypterus maculatus.
Locality: off the coastal zone of Puerto Santa Rosa (6°52´S, 178 79°55´W), Lambayeque Region, northern Peru.
Site in host: body cavity.
Specimens deposited: Hologenophore (MUSM-HEL 5142).
Representative DNA sequence:
Description …
Skrjabinisakis brevispiculata (Dollfus, 1966) ….
Lines 165,185 – one specimen. Numbers less than 10 must be written in words.
Line 205 – the same.
Lines 170, 190, 210 – Triradiate oral aperture? May be better: “Triangular oral opening”?
Lines 187, 207 – Please, use “anterior and posterior body ends” (or “parts”) instead “anterior and posterior regions”.
Line 259 – Please, rephrase sentence. I may suggest to the authors to make simpler sentences to make the text clearer.
Lines 325-327 – This information should be added to the abstract (and should be part of the Сonclusion).
The manuscript may be published in Diversity, but minor corrections are needed.

Author Response
Dear reviewer, we would like to thank you for your valuable suggestions. All comments and observations were taken into consideration. We have given our considerations in those comments with which we do not agree. Please see the attachment

Reviewer 2 Report
The Ms of Chero et al. explore the morphological and molecular characterization of Anisakid nematode larvae (Nematoda: Anisakidae) in Genypterus maculatus from the Southeastern Pacific Ocean off Peru.
This work aims to was to identify anisakid nematodes found in the black cusk-eel, a popular fish in local markets from northern Peru, using combined molecular (mitochondrial cytochrome c-oxidase subunit II) and morphological (light and scanning electron microscopy) approaches.
The manuscript is clear, relevant for the field and presented in a well-structured manner. The topic is interesting, but some aspects should be clarified.
- In M&M the authors must indicate season of catch of Genypterus maculatus specimens.
- How was the sample size selected?
- In M&M, only a previous visual inspection is indicated as a tool to obtain samples of parasites. Why an artificial enzymatic digestion not was used?
- The results obtained in this study increase the knowledge about the diversity and distribution of Anisakis and Skrjabinisakis species in the waters of the South American Pacific, however, its relevance for public health must be emphasized in the discussion, providing information to human consumers of fish.
- Accession numbers must be included.
Author Response
Dear reviewer,
We would like to thank you for your valuable suggestions. All comments and observations were taken into consideration. We have given our considerations in those comments with which we do not agree.

Reviewer 3 Report
The article is interesting. It shows the diversity of occurrence of different species within the Anisakidae family.
As for the substantive part, I have no objections.
Instead, it suggests including some additional information (in the introduction) showing the significance of the parasite. Also taking into account the current taxonomy based on genetic research may dispel some doubts. An example is the information about fish as the definitive host in the introduction. The Anisakide family are roundworms of aquatic mammals. On the other hand, the forms of the parasite found in both planktivorous and predatory fish are larval forms - therefore the fish are paratenic hosts.
In the introduction, he suggests adding important information about the threat to humans, which is of a two-fold nature. On the one hand, as a larval invasion - eating live larval forms. On the other hand, as hypersensitivity to thermostable antigens eaten even with dead larvae. For humans, this is not an example of a parasitic invasion, but of "human food fish poisoning", where the pathogenic factor is thermostable proteins of larval origin that cause hypersensitivity reactions.
Author Response
We would like to thank you for your valuable suggestions. All comments and observations were taken into consideration. We have given our considerations in those comments with which we do not agree. Please see the attachment.
